# Variance Reduced Stochastic Gradient Descent with Neighbors

**Thomas Hofmann**
Department of Computer Science
ETH Zurich, Switzerland

**Aurelien Lucchi**
Department of Computer Science
ETH Zurich, Switzerland

**Simon Lacoste-Julien**
INRIA - Sierra Project-Team
École Normale Supérieure, Paris, France

**Brian McWilliams**
Department of Computer Science
ETH Zurich, Switzerland

## Abstract

Stochastic Gradient Descent (SGD) is a workhorse in machine learning, yet its slow convergence can be a computational bottleneck. Variance reduction techniques such as SAG, SVRG and SAGA have been proposed to overcome this weakness, achieving linear convergence. However, these methods are either based on computations of full gradients at pivot points, or on keeping per data point corrections in memory. Therefore speed-ups relative to SGD may need a minimal number of epochs in order to materialize. This paper investigates algorithms that can exploit neighborhood structure in the training data to share and re-use information about past stochastic gradients across data points, which offers advantages in the transient optimization phase. As a side-product we provide a unified convergence analysis for a family of variance reduction algorithms, which we call memorization algorithms. We provide experimental results supporting our theory.

## 1 Introduction

We consider a general problem that is pervasive in machine learning, namely optimization of an empirical or regularized convex risk function. Given a convex loss $l$ and a $\mu$-strongly convex regularizer $\Omega$, one aims at finding a parameter vector $w$ which minimizes the (empirical) expectation:

$$w^* = \underset{w}{\operatorname{argmin}} f(w), \quad f(w) = \frac{1}{n}\sum_{i=1}^{n} f_i(w), \quad f_i(w) := l(w, (x_i, y_i)) + \Omega(w). \tag{1}$$

We assume throughout that each $f_i$ has $L$-Lipschitz-continuous gradients. Steepest descent can find the minimizer $w^*$, but requires repeated computations of full gradients $f'(w)$, which becomes prohibitive for massive data sets. Stochastic gradient descent (SGD) is a popular alternative, in particular in the context of large-scale learning [2, 10]. SGD updates only involve $f_i'(w)$ for an index $i$ chosen uniformly at random, providing an unbiased gradient estimate, since $\mathbf{E}f_i'(w) = f'(w)$.

It is a surprising recent finding [11, 5, 9, 6] that the finite sum structure of $f$ allows for significantly faster convergence in expectation. Instead of the standard $O(1/t)$ rate of SGD for strongly-convex functions, it is possible to obtain linear convergence with geometric rates. While SGD requires asymptotically vanishing learning rates, often chosen to be $O(1/t)$ [7], these more recent methods introduce corrections that ensure convergence for constant learning rates.

Based on the work mentioned above, the contributions of our paper are as follows: First, we define a family of variance reducing SGD algorithms, called memorization algorithms, which includes SAGA and SVRG as special cases, and develop a unifying analysis technique for it. Second, we

show geometric rates for all step sizes $\gamma < \frac{1}{4L}$, including a universal ($\mu$-independent) step size choice, providing the first $\mu$-adaptive convergence proof for SVRG. Third, based on the above analysis, we present new insights into the trade-offs between freshness and biasedness of the corrections computed from previous stochastic gradients. Fourth, we propose a new class of algorithms that resolves this trade-off by computing corrections based on stochastic gradients at neighboring points. We experimentally show its benefits in the regime of learning with a small number of epochs.

## 2 Memorization Algorithms

### 2.1 Algorithms

**Variance Reduced SGD** Given an optimization problem as in (1), we investigate a class of stochastic gradient descent algorithms that generates an iterate sequence $w^t$ ($t \geq 0$) with updates taking the form:

$$w^+ = w - \gamma g_i(w), \quad g_i(w) = f_i'(w) - \bar{\alpha}_i \quad \text{with} \quad \bar{\alpha}_i := \alpha_i - \bar{\alpha}, \tag{2}$$

where $\bar{\alpha} := \frac{1}{n} \sum_{j=1}^{n} \alpha_j$. Here $w$ is the current and $w^+$ the new parameter vector, $\gamma$ is the step size, and $i$ is an index selected uniformly at random. $\bar{\alpha}_i$ are variance correction terms such that $\mathbf{E}\bar{\alpha}_i = 0$, which guarantees unbiasedness $\mathbf{E}g_i(w) = f'(w)$. The aim is to define updates of asymptotically vanishing variance, i.e. $g_i(w) \to 0$ as $w \to w^*$, which requires $\bar{\alpha}_i \to f_i'(w^*)$. This implies that corrections need to be designed in a way to exactly cancel out the stochasticity of $f_i'(w^*)$ at the optimum. How the *memory* $\alpha_j$ is updated distinguishes the different algorithms that we consider.

**SAGA** The SAGA algorithm [4] maintains variance corrections $\alpha_i$ by memorizing stochastic gradients. The update rule is $\alpha_i^+ = f_i'(w)$ for the selected $i$, and $\alpha_j^+ = \alpha_j$, for $j \neq i$. Note that these corrections will be used the next time the same index $i$ gets sampled. Setting $\bar{\alpha}_i := \alpha_i - \bar{\alpha}$ guarantees unbiasedness. Obviously, $\bar{\alpha}$ can be updated incrementally. SAGA reuses the stochastic gradient $f_i'(w)$ computed at step $t$ to update $w$ as well as $\bar{\alpha}_i$.

**$q$-SAGA** We also consider $q$-SAGA, a method that updates $q \geq 1$ randomly chosen $\alpha_j$ variables at each iteration. This is a convenient reference point to investigate the advantages of "fresher" corrections. Note that in SAGA the corrections will be on average $n$ iterations "old". In $q$-SAGA this can be controlled to be $n/q$ at the expense of additional gradient computations.

**SVRG** We reformulate a variant of SVRG [5] in our framework using a randomization argument similar to (but simpler than) the one suggested in [6]. Fix $q > 0$ and draw in each iteration $r \sim$ Uniform$[0; 1)$. If $r < q/n$, a complete update, $\alpha_j^+ = f_j'(w)$ ($\forall j$) is performed, otherwise they are left unchanged. While $q$-SAGA updates exactly $q$ variables in each iteration, SVRG occasionally updates all $\alpha$ variables by triggering an additional sweep through the data. There is an option to not maintain $\alpha$ variables explicitly and to save on space by storing only $\bar{\alpha} = f'(w)$ and $w$.

**Uniform Memorization Algorithms** Motivated by SAGA and SVRG, we define a class of algorithms, which we call *uniform memorization algorithms*.

**Definition 1.** *A uniform $q$-memorization algorithm evolves iterates $w$ according to Eq. (2) and selects in each iteration a random index set $J$ of memory locations to update according to*

$$\alpha_j^+ := \begin{cases} f_j'(w) & \text{if } j \in J \\ \alpha_j & \text{otherwise,} \end{cases} \tag{3}$$

*such that any $j$ has the* same *probability of $q/n$ of being updated, i.e. $\forall j$, $\sum_{J \ni j} \mathbf{P}\{J\} = \frac{q}{n}$.*

Note that $q$-SAGA and the above SVRG are special cases. For $q$-SAGA: $\mathbf{P}\{J\} = 1/\binom{n}{q}$ if $|J| = q$ $\mathbf{P}\{J\} = 0$ otherwise. For SVRG: $\mathbf{P}\{\emptyset\} = 1 - q/n$, $\mathbf{P}\{[1:n]\} = q/n$, $\mathbf{P}\{J\} = 0$, otherwise.

**$\mathcal{N}$-SAGA** Because we need it in Section 3, we will also define an algorithm, which we call $\mathcal{N}$-SAGA, which makes use of a neighborhood system $\mathcal{N}_i \subseteq \{1, \ldots, n\}$ and which selects neighborhoods uniformly, i.e. $\mathbf{P}\{\mathcal{N}_i\} = \frac{1}{n}$. Note that Definition 1 requires $|\{i : j \in \mathcal{N}_i\}| = q$ ($\forall j$).

Finally, note that for generalized linear models where $f_i$ depends on $x_i$ only through $\langle w, x_i \rangle$, we get $f_i'(w) = \xi_i'(w) x_i$, i.e. the update direction is determined by $x_i$, whereas the effective step length depends on the derivative of a scalar function $\xi_i(w)$. As used in [9], this leads to significant memory savings as one only needs to store the scalars $\xi_i'(w)$ as $x_i$ is always *given* when performing an update.

## 2.2 Analysis

**Recurrence of Iterates**   The evolution equation (2) in expectation implies the recurrence (by crucially using the unbiasedness condition $\mathbf{E} g_i(w) = f'(w)$):

$$\mathbf{E}\|w^+ - w^*\|^2 = \|w - w^*\|^2 - 2\gamma \langle f'(w), w - w^* \rangle + \gamma^2 \mathbf{E}\|g_i(w)\|^2 \,. \tag{4}$$

Here and in the rest of this paper, expectations are always taken only with respect to $i$ (conditioned on the past). We utilize a number of bounds (see [4]), which exploit strong convexity of $f$ (wherever $\mu$ appears) as well as Lipschitz continuity of the $f_i$-gradients (wherever $L$ appears):

$$\langle f'(w), w - w^* \rangle \geq f(w) - f(w^*) + \tfrac{\mu}{2}\|w - w^*\|^2 \,, \tag{5}$$

$$\mathbf{E}\|g_i(w)\|^2 \leq 2\mathbf{E}\|f_i'(w) - f_i'(w^*)\|^2 + 2\mathbf{E}\|\bar{\alpha}_i - f_i'(w^*)\|^2 \,, \tag{6}$$

$$\|f_i'(w) - f_i'(w^*)\|^2 \leq 2L h_i(w), \quad h_i(w) := f_i(w) - f_i(w^*) - \langle w - w^*, f_i'(w^*) \rangle \,, \tag{7}$$

$$\mathbf{E}\|f_i'(w) - f_i'(w^*)\|^2 \leq 2L f^\delta(w), \quad f^\delta(w) := f(w) - f(w^*) \,, \tag{8}$$

$$\mathbf{E}\|\bar{\alpha}_i - f_i'(w^*)\|^2 = \mathbf{E}\|\alpha_i - f_i'(w^*)\|^2 - \|\bar{\alpha}\|^2 \leq \mathbf{E}\|\alpha_i - f_i'(w^*)\|^2. \tag{9}$$

Eq. (6) can be generalized [4] using $\|x \pm y\|^2 \leq (1+\beta)\|x\|^2 + (1+\beta^{-1})\|y\|^2$ with $\beta > 0$. However for the sake of simplicity, we sacrifice tightness and choose $\beta = 1$. Applying all of the above yields:

**Lemma 1.** *For the iterate sequence of any algorithm that evolves solutions according to Eq. (2), the following holds for a single update step, in expectation over the choice of $i$:*

$$\|w - w^*\|^2 - \mathbf{E}\|w^+ - w^*\|^2 \geq \ \gamma\mu\|w - w^*\|^2 - 2\gamma^2 \mathbf{E}\|\alpha_i - f_i'(w^*)\|^2 + \left(2\gamma - 4\gamma^2 L\right) f^\delta(w) \,.$$

All proofs are deferred to the Appendix.

**Ideal and Approximate Variance Correction**   Note that in the ideal case of $\alpha_i = f_i'(w^*)$, we would immediately get a condition for a contraction by choosing $\gamma = \frac{1}{2L}$, yielding a rate of $1 - \rho$ with $\rho = \gamma\mu = \frac{\mu}{2L}$, which is half the inverse of the condition number $\kappa := L/\mu$.

How can we further bound $\mathbf{E}\|\alpha_i - f_i'(w^*)\|^2$ in the case of "non-ideal" variance-reducing SGD? A key insight is that for memorization algorithms, we can apply the smoothness bound in Eq. (7)

$$\|\alpha_i - f_i'(w^*)\|^2 = \|f_i'(w^{\tau_i}) - f_i'(w^*)\|^2 \leq 2L h_i(w^{\tau_i}), \quad \text{(where } w^{\tau_i} \text{ is old } w\text{)}. \tag{10}$$

Note that if we only had approximations $\beta_i$ in the sense that $\|\beta_i - \alpha_i\|^2 \leq \epsilon_i$ (see Section 3), then we can use $\|x - y\| \leq 2\|x\| + 2\|y\|$ to get the somewhat worse bound:

$$\|\beta_i - f_i'(w^*)\|^2 \leq 2\|\alpha_i - f_i'(w^*)\|^2 + 2\|\beta_i - \alpha_i\|^2 \leq 4L h_i(w^{\tau_i}) + 2\epsilon_i. \tag{11}$$

**Lyapunov Function**   Ideally, we would like to show that for a suitable choice of $\gamma$, each iteration results in a contraction $\mathbf{E}\|w^+ - w^*\|^2 \leq (1-\rho)\|w - w^*\|^2$, where $0 < \rho \leq 1$. However, the main challenge arises from the fact that the quantities $\alpha_i$ represent stochastic gradients from previous iterations. This requires a somewhat more complex proof technique. Adapting the Lyapunov function method from [4], we define upper bounds $H_i \geq \|\alpha_i - f_i'(w^*)\|^2$ such that $H_i \to 0$ as $w \to w^*$. We start with $\alpha_i^0 = 0$ and (conceptually) initialize $H_i = \|f_i'(w^*)\|^2$, and then update $H_i$ in sync with $\alpha_i$,

$$H_i^+ := \begin{cases} 2L\, h_i(w) & \text{if } \alpha_i \text{ is updated} \\ H_i & \text{otherwise} \end{cases} \tag{12}$$

so that we always maintain valid bounds $\|\alpha_i - f_i'(w^*)\|^2 \leq H_i$ and $\mathbf{E}\|\alpha_i - f_i'(w^*)\|^2 \leq \bar{H}$ with $\bar{H} := \frac{1}{n}\sum_{i=1}^n H_i$. The $H_i$ are quantities showing up in the analysis, but need *not* be computed. We now define a $\sigma$-parameterized family of Lyapunov functions[1]

$$\mathcal{L}_\sigma(w, H) := \|w - w^*\|^2 + S\sigma \bar{H}, \quad \text{with } S := \left(\frac{\gamma n}{Lq}\right) \quad \text{and} \quad 0 \leq \sigma \leq 1. \tag{13}$$

In expectation under a random update, the Lyapunov function $\mathcal{L}_\sigma$ changes as $\mathbf{E}\mathcal{L}_\sigma(w^+, H^+) = \mathbf{E}\|w^+ - w^*\|^2 + S\sigma\,\mathbf{E}\bar{H}^+$. We can readily apply Lemma 1 to bound the first part. The second part is due to (12), which mirrors the update of the $\alpha$ variables. By crucially using the property that any $\alpha_j$ has the same probability of being updated in (3), we get the following result:

**Lemma 2.** *For a uniform $q$-memorization algorithm, it holds that*

$$\mathbf{E}\bar{H}^+ = \left(\frac{n-q}{n}\right)\bar{H} + \frac{2Lq}{n}\,f^\delta(w). \tag{14}$$

Note that in expectation the shrinkage does not depend on the location of previous iterates $w^\tau$ and the new increment is proportional to the sub-optimality of the current iterate $w$. Technically, this is how the possibly complicated dependency on previous iterates is dealt with in an effective manner.

**Convergence Analysis**  We first state our main Lemma about Lyapunov function contractions:

**Lemma 3.** *Fix $c \in (0;1]$ and $\sigma \in [0;1]$ arbitrarily. For any uniform $q$-memorization algorithm with sufficiently small step size $\gamma$ such that*

$$\gamma \le \frac{1}{2L}\min\left\{\frac{K\sigma}{K+2c\sigma}, 1-\sigma\right\}, \quad and \quad K := \frac{4qL}{n\mu}, \tag{15}$$

*we have that*

$$\mathbf{E}\mathcal{L}_\sigma(w^+, H^+) \le (1-\rho)\mathcal{L}_\sigma(w, H), \quad with \quad \rho := c\mu\gamma. \tag{16}$$

*Note that $\gamma < \frac{1}{2L}\max_{\sigma\in[0,1]}\min\{\sigma, 1-\sigma\} = \frac{1}{4L}$ (in the $c \to 0$ limit).*

By maximizing the bounds in Lemma 3 over the choices of $c$ and $\sigma$, we obtain our main result that provides guaranteed geometric rates for all step sizes up to $\frac{1}{4L}$.

**Theorem 1.** *Consider a uniform $q$-memorization algorithm. For any step size $\gamma = \frac{a}{4L}$ with $a < 1$, the algorithm converges at a geometric rate of at least $(1-\rho(\gamma))$ with*

$$\rho(\gamma) = \frac{q}{n}\cdot\frac{1-a}{1-a/2} = \frac{\mu}{4L}\cdot\frac{K(1-a)}{1-a/2}, \quad if\ \gamma \ge \gamma^*(K), \quad otherwise\ \ \rho(\gamma) = \mu\gamma \tag{17}$$

*where*

$$\gamma^*(K) := \frac{a^*(K)}{4L}, \quad a^*(K) := \frac{2K}{1+K+\sqrt{1+K^2}}, \quad K := \frac{4qL}{n\mu} = \frac{4q}{n}\kappa. \tag{18}$$

We would like to provide more insights into this result.

**Corollary 1.** *In Theorem 1, $\rho$ is maximized for $\gamma = \gamma^*(K)$. We can write $\rho^*(K) = \rho(\gamma^*)$ as*

$$\rho^*(K) = \frac{\mu}{4L}a^*(K) = \frac{q}{n}\frac{a^*(K)}{K} = \frac{q}{n}\left[\frac{2}{1+K+\sqrt{1+K^2}}\right] \tag{19}$$

*In the big data regime $\rho^* = \frac{q}{n}(1-\frac{1}{2}K+O(K^3))$, whereas in the ill-conditioned case $\rho^* = \frac{\mu}{4L}(1-\frac{1}{2}K^{-1}+O(K^{-3}))$.*

The guaranteed rate is bounded by $\frac{\mu}{4L}$ in the regime where the condition number dominates $n$ (large $K$) and by $\frac{q}{n}$ in the opposite regime of large data (small $K$). Note that if $K \le 1$, we have $\rho^* = \zeta\frac{q}{n}$ with $\zeta \in [2/(2+\sqrt{2});1] \approx [0.585;1]$. So for $q \le n\frac{\mu}{4L}$, it pays off to increase freshness as it affects the rate proportionally. In the ill-conditioned regime ($\kappa > n$), the influence of $q$ vanishes.

Note that for $\gamma \ge \gamma^*(K)$, $\gamma \to \frac{1}{4L}$ the rate decreases monotonically, yet the decrease is only minor. With the exception of a small neighborhood around $\frac{1}{4L}$, the entire range of $\gamma \in [\gamma^*;\frac{1}{4L})$ results in very similar rates. Underestimating $\gamma^*$ however leads to a (significant) slow-down by a factor $\gamma/\gamma^*$.

As the optimal choice of $\gamma$ depends on $K$, i.e. $\mu$, we would prefer step sizes that are $\mu$-independent, thus giving rates that adapt to the local curvature (see [9]). It turns out that by choosing a step size that maximizes $\min_K \rho(\gamma)/\rho^*(K)$, we obtain a $K$-agnostic step size with rate off by at most $1/2$:

**Corollary 2.** *Choosing $\gamma = \frac{2-\sqrt{2}}{4L}$, leads to $\rho(\gamma) \ge (2-\sqrt{2})\rho^*(K) > \frac{1}{2}\rho^*(K)$ for all $K$.*

To gain more insights into the trade-offs for these fixed large universal step sizes, the following corollary details the range of rates obtained:

**Corollary 3.** *Choosing $\gamma = \frac{a}{4L}$ with $a < 1$ yields $\rho = \min\{\frac{1-a}{1-\frac{1}{2}a}\frac{q}{n}, \frac{a}{4}\frac{\mu}{L}\}$. In particular, we have for the choice $\gamma = \frac{1}{5L}$ that $\rho = \min\{\frac{1}{3}\frac{q}{n}, \frac{1}{5}\frac{\mu}{L}\}$ (roughly matching the rate given in [4] for $q = 1$).*

## 3 Sharing Gradient Memory

### 3.1 $\epsilon$-Approximation Analysis

As we have seen, fresher gradient memory, i.e. a larger choice for $q$, affects the guaranteed convergence rate as $\rho \sim q/n$. However, as long as one step of a $q$-memorization algorithm is as expensive as $q$ steps of a 1-memorization algorithm, this insight does not lead to practical improvements *per se*. Yet, it raises the question, whether we can accelerate these methods, in particular $\mathcal{N}$-SAGA, by approximating gradients stored in the $\alpha_i$ variables. Note that we are always using the correct stochastic gradients in the *current* update and by assuring $\sum_i \bar{\alpha}_i = 0$, we will not introduce any bias in the update direction. Rather, we lose the guarantee of asymptotically vanishing variance at $w^*$. However, as we will show, it is possible to retain geometric rates up to a $\delta$-ball around $w^*$.

We will focus on SAGA-style updates for concreteness and investigate an algorithm that mirrors $\mathcal{N}$-SAGA with the only difference that it maintains approximations $\beta_i$ to the true $\alpha_i$ variables. We aim to guarantee $\mathbf{E}\|\alpha_i - \beta_i\|^2 \leq \epsilon$ and will use Eq. (11) to modify the right-hand-side of Lemma 1. We see that approximation errors $\epsilon_i$ are multiplied with $\gamma^2$, which implies that we should aim for small learning rates, ideally without compromising the $\mathcal{N}$-SAGA rate. From Theorem 1 and Corollary 1 we can see that we can choose $\gamma \lesssim q/\mu n$ for $n$ sufficiently large, which indicates that there is hope to dampen the effects of the approximations. We now make this argument more precise.

**Theorem 2.** *Consider a uniform $q$-memorization algorithm with $\alpha$-updates that are on average $\epsilon$-accurate (i.e. $\mathbf{E}\|\alpha_i - \beta_i\|^2 \leq \epsilon$). For any step size $\gamma \leq \tilde{\gamma}(K)$, where $\tilde{\gamma}$ is given by Corollary 5 in the appendix (note that $\tilde{\gamma}(K) \geq \frac{2}{3}\gamma^*(K)$ and $\tilde{\gamma}(K) \to \gamma^*(K)$ as $K \to 0$), we get*

$$\mathbb{E}\mathcal{L}(w^t, H^t) \leq (1 - \mu\gamma)^t \mathcal{L}_0 + \frac{4\gamma\epsilon}{\mu}, \quad \text{with } \mathcal{L}_0 := \|w^0 - w^*\|^2 + s(\gamma)\mathbf{E}\|f_i(w^*)\|^2, \quad (20)$$

*where $\mathbb{E}$ denote the (unconditional) expectation over histories (in contrast to $\mathbf{E}$ which is conditional), and $s(\gamma) := \frac{4\gamma}{K\mu}(1 - 2L\gamma)$.*

**Corollary 4.** *With $\gamma = \min\{\mu, \tilde{\gamma}(K)\}$ we have*

$$\frac{4\gamma\epsilon}{\mu} \leq 4\epsilon, \quad \text{with a rate} \quad \rho = \min\{\mu^2, \mu\tilde{\gamma}\}. \quad (21)$$

In the relevant case of $\mu \sim 1/\sqrt{n}$, we thus converge towards some $\sqrt{\epsilon}$-ball around $w^*$ at a similar rate as for the exact method. For $\mu \sim n^{-1}$, we have to reduce the step size significantly to compensate the extra variance and to still converge to an $\sqrt{\epsilon}$-ball, resulting in the slower rate $\rho \sim n^{-2}$, instead of $\rho \sim n^{-1}$.

We also note that the geometric convergence of SGD with a constant step size to a neighborhood of the solution (also proven in [8]) can arise as a special case in our analysis. By setting $\alpha_i = 0$ in Lemma 1, we can take $\epsilon = \mathbf{E}\|f_i'(w^*)\|^2$ for SGD. An approximate $q$-memorization algorithm can thus be interpreted as making $\epsilon$ an algorithmic parameter, rather than a fixed value as in SGD.

### 3.2 Algorithms

**Sharing Gradient Memory**  We now discuss our proposal of using neighborhoods for sharing gradient information between close-by data points. Thereby we avoid an increase in gradient computations relative to $q$- or $\mathcal{N}$-SAGA at the expense of suffering an approximation bias. This leads to a new tradeoff between freshness and approximation quality, which can be resolved in non-trivial ways, depending on the desired final optimization accuracy.

We distinguish two types of quantities. First, the gradient memory $\alpha_i$ as defined by the reference algorithm $\mathcal{N}$-SAGA. Second, the shared gradient memory state $\beta_i$, which is used in a modified update rule in Eq. (2), i.e. $w^+ = w - \gamma(f_i'(w) - \beta_i + \bar{\beta})$. Assume that we select an index $i$ for the weight update, then we generalize Eq. (3) as follows

$$\beta_j^+ := \begin{cases} f_i'(w) & \text{if } j \in \mathcal{N}_i \\ \beta_j & \text{otherwise} \end{cases}, \quad \bar{\beta} := \frac{1}{n}\sum_{i=1}^n \beta_i, \quad \bar{\beta}_i := \beta_i - \bar{\beta}. \quad (22)$$

In the important case of generalized linear models, where one has $f_i'(w) = \xi_i'(w)x_i$, we can modify the relevant case in Eq. (22) by $\beta_j^+ := \xi_i'(w)x_j$. This has the advantages of using the correct direction, while reducing storage requirements.

**Approximation Bounds**  For our analysis, we need to control the error $\|\alpha_i - \beta_i\|^2 \le \epsilon_i$. This obviously requires problem-specific investigations.

Let us first look at the case of ridge regression. $f_i(w) := \frac{1}{2}(\langle x_i, w \rangle - y_i)^2 + \frac{\lambda}{2}\|w\|^2$ and thus $f_i'(w) = \xi_i'(w)x_i + \lambda w$ with $\xi_i'(w) := \langle x_i, w \rangle - y_i$. Considering $j \in \mathcal{N}_i$ being updated, we have

$$\|\alpha_j^+ - \beta_j^+\| = |\xi_j'(w) - \xi_i'(w)| \|x_j\| \le (\delta_{ij}\|w\| + |y_j - y_i|) \|x_j\| =: \epsilon_{ij}(w) \tag{23}$$

where $\delta_{ij} := \|x_i - x_j\|$. Note that this can be pre-computed with the exception of the norm $\|w\|$ that we only know at the time of an update.

Similarly, for regularized logistic regression with $y \in \{-1, 1\}$, we have $\xi_i'(w) = y_i/(1 + e^{y_i \langle x_i, w \rangle})$. With the requirement on neighbors that $y_i = y_j$ we get

$$\|\alpha_j^+ - \beta_j^+\| \le \frac{e^{\delta_{ij}\|w\|} - 1}{1 + e^{-\langle x_i, w \rangle}} \|x_j\| =: \epsilon_{ij}(w) \tag{24}$$

Again, we can pre-compute $\delta_{ij}$ and $\|x_j\|$. In addition to $\xi_i'(w)$ we can also store $\langle x_i, w \rangle$.

**$\epsilon\mathcal{N}$-SAGA**  We can use these bounds in two ways. First, assuming that the iterates stay within a norm-ball (e.g. $L_2$-ball), we can derive upper bounds

$$\epsilon_j(r) \ge \max\{\epsilon_{ij}(w) : j \in \mathcal{N}_i, \|w\| \le r\}, \qquad \epsilon(r) = \frac{1}{n}\sum_j \epsilon_j(r). \tag{25}$$

Obviously, the more compact the neighborhoods are, the smaller $\epsilon(r)$. This is most useful for the analysis. Second, we can specify a target accuracy $\epsilon$ and then prune neighborhoods dynamically. This approach is more practically relevant as it allows us to directly control $\epsilon$. However, a dynamically varying neighborhood violates Definition 1. We fix this in a sound manner by modifying the memory updates as follows:

$$\beta_j^+ := \begin{cases} f_i'(w) & \text{if } j \in \mathcal{N}_i \text{ and } \epsilon_{ij}(w) \le \epsilon \\ f_j'(w) & \text{if } j \in \mathcal{N}_i \text{ and } \epsilon_{ij}(w) > \epsilon \\ \beta_j & \text{otherwise} \end{cases} \tag{26}$$

This allows us to interpolate between sharing more aggressively (saving computation) and performing more computations in an exact manner. In the limit of $\epsilon \to 0$, we recover $\mathcal{N}$-SAGA, as $\epsilon \to \epsilon^{\max}$ we recover the first variant mentioned.

**Computing Neighborhoods**  Note that the pairwise Euclidean distances show up in the bounds in Eq. (23) and (24). In the classification case we also require $y_i = y_j$, whereas in the ridge regression case, we also want $|y_i - y_j|$ to be small. Thus modulo filtering, this suggests the use of Euclidean distances as the metric for defining neighborhoods. Standard approximation techniques for finding near(est) neighbors can be used. This comes with a computational overhead, yet the additional costs will amortize over multiple runs or multiple data analysis tasks.

## 4  Experimental Results

**Algorithms**  We present experimental results on the performance of the different variants of memorization algorithms for variance reduced SGD as discussed in this paper. SAGA has been uniformly superior to SVRG in our experiments, so we compare SAGA and $\epsilon\mathcal{N}$-SAGA (from Eq. (26)), alongside with SGD as a straw man and $q$-SAGA as a point of reference for speed-ups. We have chosen $q = 20$ for $q$-SAGA and $\epsilon\mathcal{N}$-SAGA. The same setting was used across all data sets and experiments.

**Data Sets**  As special cases for the choice of the loss function and regularizer in Eq. (1), we consider two commonly occurring problems in machine learning, namely least-square regression and $\ell_2$-regularized logistic regression. We apply least-square regression on the million song year regression from the UCI repository. This dataset contains $n = 515,345$ data points, each described by $d = 90$ input features. We apply logistic regression on the *cov* and *ijcnn1* datasets obtained from the *libsvm* website [2]. The *cov* dataset contains $n = 581,012$ data points, each described by $d = 54$ input features. The *ijcnn1* dataset contains $n = 49,990$ data points, each described by $d = 22$ input features. We added an $\ell_2$-regularizer $\Omega(w) = \mu\|w\|_2^2$ to ensure the objective is strongly convex.

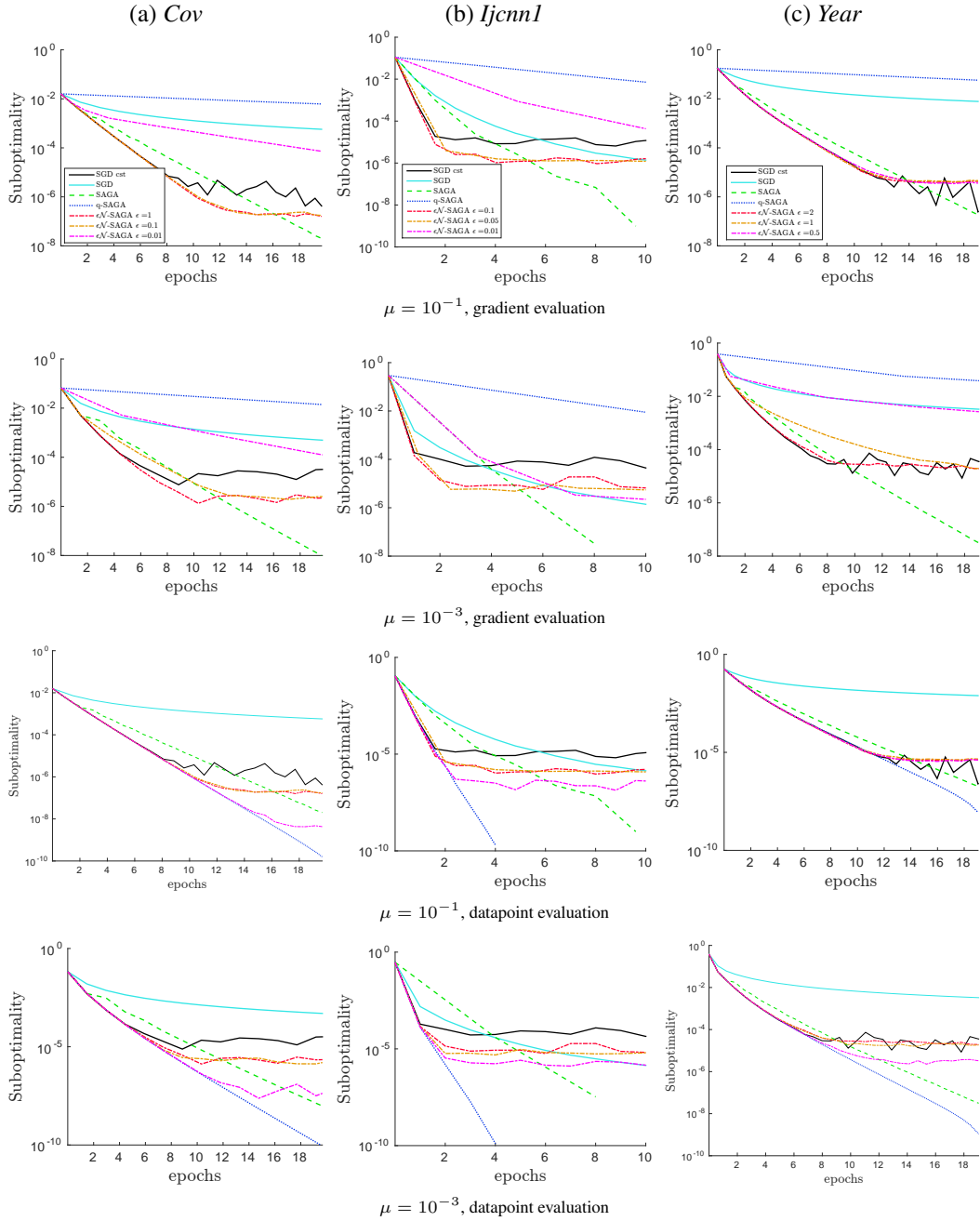

Figure 1: Comparison of $\epsilon\mathcal{N}$-SAGA, $q$-SAGA, SAGA and SGD (with decreasing and constant step size) on three datasets. The top two rows show the suboptimality as a function of the number of gradient evaluations for two different values of $\mu = 10^{-1}, 10^{-3}$. The bottom two rows show the suboptimality as a function of the number of datapoint evaluations (i.e. number of stochastic updates) for two different values of $\mu = 10^{-1}, 10^{-3}$.

**Experimental Protocol**  We have run the algorithms in question in an i.i.d. sampling setting and averaged the results over 5 runs. Figure 1 shows the evolution of the suboptimality $f^\delta$ of the objective as a function of two different metrics: (1) in terms of the number of update steps performed ("datapoint evaluation"), and (2) in terms of the number of gradient computations ("gradient evaluation"). Note that SGD and SAGA compute one stochastic gradient per update step unlike $q$-SAGA, which is included here not as a practically relevant algorithm, but as an indication of potential improvements that could be achieved by fresher corrections. A step size $\gamma = \frac{q}{\mu n}$ was used everywhere, except for "plain SGD". Note that as $K \ll 1$ in all cases, this is close to the optimal value suggested by our analysis; moreover, using a step size of $\sim \frac{1}{L}$ for SAGA as suggested in previous work [9] did not appear to give better results. For plain SGD, we used a schedule of the form $\gamma_t = \gamma_0/t$ with constants optimized coarsely via cross-validation. The $x$-axis is expressed in units of $n$ (suggestively called "epochs").

**SAGA vs. SGD cst**  As we can see, if we run SGD with the same constant step size as SAGA, it takes several epochs until SAGA really shows a significant gain. The constant step-size variant of SGD is faster in the early stages until it converges to a neighborhood of the optimum, where individual runs start showing a very noisy behavior.

**SAGA vs. $q$-SAGA**  $q$-SAGA outperforms plain SAGA quite consistently when counting stochastic update steps. This establishes optimistic reference curves of what we can expect to achieve with $\epsilon\mathcal{N}$-SAGA. The actual speed-up is somewhat data set dependent.

**$\epsilon\mathcal{N}$-SAGA vs. SAGA and $q$-SAGA**  $\epsilon\mathcal{N}$-SAGA with sufficiently small $\epsilon$ can realize much of the possible freshness gains of $q$-SAGA and performs very similar for a few (2-10) epochs, where it traces nicely between the SAGA and $q$-SAGA curves. We see solid speed-ups on all three datasets for both $\mu = 0.1$ and $\mu = 0.001$.

**Asymptotics**  It should be clearly stated that running $\epsilon\mathcal{N}$-SAGA at a fixed $\epsilon$ for longer will not result in good asymptotics on the empirical risk. This is because, as theory predicts, $\epsilon\mathcal{N}$-SAGA can not drive the suboptimality to zero, but rather levels-off at a point determined by $\epsilon$. In our experiments, the cross-over point with SAGA was typically after $5 - 15$ epochs. Note that the gains in the first epochs can be significant, though. In practice, one will either define a desired accuracy level and choose $\epsilon$ accordingly or one will switch to SAGA for accurate convergence.

## 5  Conclusion

We have generalized variance reduced SGD methods under the name of memorization algorithms and presented a corresponding analysis, which commonly applies to all such methods. We have investigated in detail the range of safe step sizes with their corresponding geometric rates as guaranteed by our theory. This has delivered a number of new insights, for instance about the trade-offs between small ($\sim \frac{1}{n}$) and large ($\sim \frac{1}{4L}$) step sizes in different regimes as well as about the role of the freshness of stochastic gradients evaluated at past iterates.

We have also investigated and quantified the effect of additional errors in the variance correction terms on the convergence behavior. Dependent on how $\mu$ scales with $n$, we have shown that such errors can be tolerated, yet, for small $\mu$, may have a negative effect on the convergence rate as much smaller step sizes are needed to still guarantee convergence to a small region. We believe this result to be relevant for a number of approximation techniques in the context of variance reduced SGD.

Motivated by these insights and results of our analysis, we have proposed $\epsilon\mathcal{N}$-SAGA, a modification of SAGA that exploits similarities between training data points by defining a neighborhood system. Approximate versions of per-data point gradients are then computed by sharing information among neighbors. This opens-up the possibility of variance-reduction in a streaming data setting, where each data point is only seen once. We believe this to be a promising direction for future work.

Empirically, we have been able to achieve consistent speed-ups for the initial phase of regularized risk minimization. This shows that approximate computations of variance correction terms constitutes a promising approach of trading-off computation with solution accuracy.

**Acknowledgments**  We would like to thank Yannic Kilcher, Martin Jaggi, Rémi Leblond and the anonymous reviewers for helpful suggestions and corrections.

## Footnotes

[1]This is a simplified version of the one appearing in [4], as we assume $f'(w^*) = 0$ (unconstrained regime).

[2] http://www.csie.ntu.edu.tw/~cjlin/libsvmtools/datasets

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
