[Supplementary Material]

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

# A Appendix

**Lemma 1.** *For the iterate sequence of any algorithm that evolves solutions according to Eq.* (2)*, the following holds for a single update step, in expectation over the choice of $i$, with $\triangle := \|w - w^*\|^2 - \mathbf{E}\|w^+ - w^*\|^2$, then:*

$$\triangle \geq \quad \gamma\mu\|w - w^*\|^2 - 2\gamma^2\mathbf{E}\|\alpha_i - f_i'(w^*)\|^2 + \left(2\gamma - 4\gamma^2 L\right) f^\delta(w).$$

*Proof.* Starting from Eq. (4) we have

$$
\begin{aligned}
\triangle &= 2\gamma\langle f'(w), w - w^*\rangle - \gamma^2\mathbf{E}\|g_i(w)\|^2 \\
&\overset{(5)}{\geq} \gamma\mu\|w - w^*\|^2 + 2\gamma f^\delta(w) - \gamma^2\mathbf{E}\|g_i(w)\|^2 \\
&\overset{(6)}{\geq} \gamma\mu\|w - w^*\|^2 + 2\gamma f^\delta(w) - 2\gamma^2\mathbf{E}\|f_i'(w) - f_i'(w^*)\|^2 - 2\gamma^2\mathbf{E}\|\bar{\alpha}_i - f_i'(w^*)\|^2 \\
&\overset{(8),(9)}{\geq} \gamma\mu\|w - w^*\|^2 + 2\gamma(1 - 2\gamma L)f^\delta(w) - 2\gamma^2\mathbf{E}\|\alpha_i - f_i'(w^*)\|^2.
\end{aligned}
$$

$\square$

**Lemma 2.** *For a uniform $q$-memorization algorithm, it holds that*

$$\mathbf{E}\bar{H}^+ = \left(\frac{n - q}{n}\right)\bar{H} + \frac{2Lq}{n} f^\delta(w).$$

*Proof.* From the uniformity property $(*)$ in Definition 1, it follows that

$$n\mathbf{E}H^+ = \sum_{i=1}^n \mathbf{E}H_i^+ \overset{(*)}{=} \sum_{i=1}^n \left(\left(1 - \frac{q}{n}\right)H_i + 2L\left(\frac{q}{n}\right)h_i(w)\right) = (n - q)\bar{H} + \frac{2Lq}{n}\sum_{i=1}^n h_i(w).$$

Exploiting the fact that $\frac{1}{n}\sum_{i=1}^n h_i(w) = f(w) - f(w^*) + 0 = f^\delta(w)$ completes the proof. $\square$

**Lemma 3.** *Fix $c \in (0; 1]$ and $\sigma \in [0; 1]$ arbitrarily. For any uniform $q$-memorization algorithm with sufficiently small step size $\gamma$ such that*

$$\gamma \leq \frac{1}{L}\min\left\{\frac{K\sigma}{2K + 4c\sigma}, \frac{1 - \sigma}{2}\right\}, \quad and \quad K := \frac{4qL}{n\mu},$$

*we have that*

$$\mathbf{E}\mathcal{L}_\sigma(w^+, H^+) \leq (1 - \rho)\mathcal{L}_\sigma(w, H), \quad with \quad \rho := c\mu\gamma. \tag{27}$$

*Note that $\gamma < \frac{1}{2L}\max_{\sigma\in[0,1]}\min\{\sigma, 1 - \sigma\} = \frac{1}{4L}$ (in the $c \to 0$ limit).*

*Proof.* From Lemma 1, we can see that we will have $\rho \leq \gamma\mu$ based on the $\|w - w^*\|^2$ part of $\mathcal{L}_\sigma$. Hence, we can write the rate as $\rho = c\mu\gamma$, where $0 < c \leq 1$.

Let us now apply both, Lemma 1 and Lemma 2, to quantify the progress guaranteed to be made in one iteration of the algorithm in expectation, combining the changes to the iterate $w \to w^+$ as well as those to the memory $\alpha \to \alpha^+$ into $\mathcal{L}_\sigma$. Set $\triangle_\sigma := \mathcal{L}_\sigma(w, H) - \mathbf{E}\mathcal{L}_\sigma^+(w, H)$, then

$$
\begin{aligned}
\triangle_\sigma &= \|w - w^*\|^2 - \mathbf{E}\|w^+ - w^*\|^2 + S\sigma\left(\bar{H} - \mathbf{E}\bar{H}^+\right) \tag{28} \\
&\geq \gamma\mu\|w - w^*\|^2 - 2\gamma^2\mathbf{E}\|\alpha_i - f_i'(w^*)\|^2 + 2\gamma\left(1 - 2\gamma L\right)f^\delta(w) \\
&\quad + S\sigma\left(\bar{H} - \left(\frac{n - q}{n}\right)\bar{H} - \frac{2Lq}{n}f^\delta(w))\right).
\end{aligned}
$$

As we argued after Eq. (12), the definition of $H_i$ combined with property (10) ensure the crucial bound $\mathbf{E}\|\alpha_i - f_i'(w^*)\|^2 \leq \bar{H}$. Including it and gathering terms in the same "units", we get:

$$\triangle_\sigma \geq \gamma\mu\|w - w^*\|^2 + \left[S\sigma\left(\frac{q}{n}\right) - 2\gamma^2\right]\bar{H} + 2\left[-S\sigma L\left(\frac{q}{n}\right) + \gamma(1 - 2\gamma L)\right]f^\delta(w) \tag{29}$$

We can further simplify the term in the second rectangular brackets with the definition of $S$ (in hindsight motivating its definition):

$$-2S\sigma L\left(\frac{q}{n}\right) + 2\gamma\left(1 - 2\gamma L\right) = 2\gamma\left[-\sigma + (1 - 2\gamma L)\right] \tag{30}$$

We require this term to be non-negative, so that we can safely drop it. This leads an upper bound requirement on the step size:

$$-\sigma + 1 - 2\gamma L \geq 0 \iff \gamma \leq \frac{1 - \sigma}{2L}. \tag{31}$$

The term in the first rectangular brackets in Eq. (29) needs to be $\geq \rho S\sigma$ in order to recover $\rho\mathcal{L}_\sigma = \rho\left(\|w - w^*\|^2 + S\sigma\bar{H}\right)$. Inserting the definition of $S$, $\rho$ and dividing by $\gamma$ yields

$$\frac{\sigma}{L} - 2\gamma \geq \frac{\rho S\sigma}{\gamma} = c\gamma\mu\frac{n\sigma}{Lq} = \frac{4c\sigma\gamma}{K} \iff \gamma \leq \frac{1}{L}\frac{K\sigma}{2K + 4c\sigma} \tag{32}$$

We can summarize the derivation in the claimed combined inequality. $\qquad\square$

**Theorem 1.** *Consider a uniform $q$-memorization algorithm. For any step size $\gamma = \frac{a}{4L}$, with $a < 1$ the algorithm converges at a geometric rate of at least $(1 - \rho(\gamma))$ with*

$$\rho(\gamma) = \frac{q}{n}\cdot\frac{1 - a}{1 - a/2} = \frac{\mu}{4L}\cdot\frac{K(1 - a)}{1 - a/2}, \quad \textit{if } \gamma \geq \gamma^*(K), \quad \textit{otherwise } \rho(\gamma) = \mu\gamma$$

*where*

$$\gamma^*(K) := \frac{a^*(K)}{4L}, \quad a^*(K) := \frac{2K}{1 + K + \sqrt{1 + K^2}}, \quad K := \frac{4qL}{n\mu}$$

*Proof.* Consider a fixed $\gamma < \frac{1}{4L}$. There are potentially (infinitely) many choices of $(c, \sigma)$ that fulfill the condition in Eq. (15). Among those, the largest rate is obtained by maximizing $c \leq 1$ as $\rho(\gamma) = c\mu\gamma$. Note that for any $\gamma$ that does not achieve Eq. (15) with equality for both terms, one can find a larger $\gamma$ with the same choice of $c$ by either increasing (slack in the first inequality) or decreasing (slack in the second inequality) $\sigma$. We thus focus on step sizes that are maximal for some choice of $(c, \sigma)$. Equality with the second bound directly gives us

$$\frac{1}{L}\frac{1 - \sigma}{2} \stackrel{!}{=} \gamma \implies \sigma^* = 1 - 2L\gamma. \tag{33}$$

We plug this into the first bound and again equal $\gamma$, which yields an optimality condition for $c$

$$L\gamma \stackrel{!}{=} \frac{K\sigma^*}{2K + 4c\sigma^*} \iff c^* = \frac{K}{4\gamma L}\left[1 - \frac{2\gamma L}{\sigma^*}\right] \implies c^* = \frac{K}{4\gamma L}\frac{1 - 4L\gamma}{1 - 2L\gamma} \tag{34}$$

and thus

$$\rho = c\mu\gamma = \frac{\mu K}{4L}\frac{1 - 4L\gamma}{1 - 2L\gamma} = \frac{q}{n}\frac{1 - 4L\gamma}{1 - 2L\gamma} \tag{35}$$

It remains to check what the admissible range of $\gamma$ is that achieves the bound in Eq. (15) as we required. The latter is determined by the constraints $c \in (0; 1]$. From Eq. (34) we can read off for $c^* > 0$,

$$1 - 4L\gamma > 0 \iff \gamma < \frac{1}{4L}. \tag{36}$$

At the other extreme of $c = 1$ we can solve the resulting quadratic equation in $\gamma$

$$\gamma = \frac{q}{n\mu}\frac{1 - 4L\gamma}{1 - 2L\gamma} = \frac{K}{4L}\frac{1 - 4L\gamma}{1 - 2L\gamma} \tag{37}$$

to get $\gamma = \gamma^*(K)$ as claimed in Eq. (18) (excluding the second root which yields $\gamma > \frac{1}{4L}$). Moreover, for $\gamma < \gamma^*(K)$ we choose $c = 1$ to maximize the rate and have $\rho = \mu\gamma$. $\qquad\square$

**Corollary 1.** *In Theorem 1, $\rho$ is maximized for $\gamma = \gamma^*(K)$. We can write $\rho^*(K) = \rho(\gamma^*)$ as*

$$\rho^*(K) = \frac{\mu}{4L}a^*(K) = \frac{q}{n}\frac{a^*(K)}{K} = \frac{q}{n}\left[\frac{2}{1+K+\sqrt{1+K^2}}\right]$$

*In the big data regime $\rho^* = \frac{q}{n}(1 - \frac{1}{2}K + O(K^3))$, whereas in the ill-conditioned case $\rho^* = \frac{\mu}{4L}(1 - \frac{1}{2}K^{-1} + O(K^{-3}))$.*

*Proof.* Plugging in the definitions of $\gamma^*(K)$ and $K$ and performing some symbolic simplifications yields the result. $\square$

**Corollary 2.** *Choosing $\gamma = \frac{2-\sqrt{2}}{4L}$, leads to $\rho(\gamma) \geq (2 - \sqrt{2})\rho^* > \frac{1}{2}\rho^*$.*

*Proof.* Write $\gamma = \frac{a}{4L}$, then if $\gamma \geq \gamma^*(K)$: $\frac{\rho}{\rho^*} \geq \frac{1-a}{1-a/2}$, otherwise: $\frac{\rho}{\rho^*} = \frac{\gamma}{\gamma^*} \geq a$, with equality when $K = \infty$. Setting both equal yields $a = 2 - \sqrt{2} \approx 0.5858$. $\square$

**Corollary 3.** *Choosing $\gamma = \frac{a}{4L}$ with $a < 1$ yields $\rho = \min\{\frac{1-a}{1-\frac{1}{2}a}\frac{q}{n}, \frac{a}{4}\frac{\mu}{L}\}$. In particular, we have for the choice $\gamma = \frac{1}{5L}$ that $\rho = \min\{\frac{1}{3}\frac{q}{n}, \frac{1}{5}\frac{\mu}{L}\}$.*

*Proof.* If $\gamma \geq \gamma^*(R)$ then $\rho = \frac{1-a}{1-a/2}\frac{q}{n} = \frac{1}{3}\frac{q}{n}$; otherwise, $\rho = \mu\gamma = \mu\frac{a}{4L} = \frac{1}{5}\frac{\mu}{L}$. $\square$

**Theorem 2.** *Consider a uniform q-memorization algorithm with $\alpha$-updates that are on average $\epsilon$-accurate (i.e. $\mathbf{E}\|\alpha_i - \beta_i\|^2 \leq \epsilon$). For any step size $\gamma \leq \tilde{\gamma}(K)$, where $\tilde{\gamma}$ is given in Eq. (39) in Corollary 5 below (note that $\tilde{\gamma}(K) \geq \frac{2}{3}\gamma^*(K)$ and $\tilde{\gamma}(K) \to \gamma^*(K)$ as $K \to 0$), we get*

$$\mathbb{E}\mathcal{L}(w^t, H^t) \leq (1 - \mu\gamma)^t \mathcal{L}_0 + \frac{4\gamma\epsilon}{\mu}, \quad \text{with } \mathcal{L}_0 := \|w^0 - w^*\|^2 + s(\gamma)\mathbf{E}\|f_i(w^*)\|^2, \qquad (38)$$

*where $\mathbb{E}$ denote the (unconditional) expectation over histories (in contrast to $\mathbf{E}$ which is conditional), and $s(\gamma) := \frac{4\gamma}{K\mu}(1 - 2L\gamma)$.*

*Proof.* Following the same line of argument as in Lemma 3 and Theorem 1 with the modifications summarized in Corollary 5

$$\mathbf{E}\mathcal{L}(w^+, H^+) \leq (1 - \gamma\mu)\mathcal{L}(w, H) + 4\gamma^2\epsilon$$

and unrolling the recurrence over $t$

$$\mathbb{E}\mathcal{L}(w^t, H^t) \leq (1 - \gamma\mu)^t \mathcal{L}(w^0, H^0) + \overbrace{\left[\sum_{s=0}^{t-1}(1 - \gamma\mu)^s\right]}^{\leq 1/(\gamma\mu)} 4\gamma^2\epsilon$$

using $\frac{1}{1-x} = \sum_{s=0}^{\infty} x^s$ applied with $x = (1 - \rho)$ (see [8] for its use for constant step size SGD). According to Eq. (13), $\mathcal{L}(w^0, H^0) = \|w - w^*\|^2 + S\sigma\bar{H}^0$, where $S = \frac{\gamma n}{Lq} = \frac{4\gamma}{K\mu}$. As the algorithm initializes $\alpha_i^0$ to 0, we have $\bar{H}^0 = \mathbf{E}\|f_i'(w^*)\|^2$. Finally, the proof of Corollary 5 follows the proof of Theorem 1 and also gives $\sigma^* = 1 - 2L\gamma$ as in Eq. (33). Substituting in $\mathcal{L}(w^0, H^0)$ gives $\mathcal{L}_0$. $\square$

**Corollary 4.** *With $\gamma = \min\{\mu, \tilde{\gamma}(K)\}$ we have*

$$\frac{4\gamma\epsilon}{\mu} \leq 4\epsilon, \qquad \text{with a rate} \quad \rho = \min\{\mu^2, \mu\tilde{\gamma}\}.$$

*Proof.* By definition, $\gamma \leq \mu$, thus $\gamma/\mu \leq 1$, yielding the first claim. By definition, we also have $\gamma \leq \tilde{\gamma}(K)$, thus from Theorem 1 adapted to Corollary 5, we know that $\rho = \mu\gamma$, which concludes the proof. (Note that with $\gamma > \tilde{\gamma}(K)$ we will increase the error, while decreasing $\rho(\gamma) \leq \tilde{\rho} = \mu\tilde{\gamma}$. This is why this choice is not sensible according to our theory.) $\square$

**Corollary 5** (Patch-ups)*. Using Eq.(11) instead of Eq. (10), but then setting $\epsilon_i = 0$ yields the same results as before with the following changes:*

*(a) Lemma 3: the bound becomes $\gamma \leq \frac{1}{L} \min\{\frac{1}{4}\frac{K\sigma}{K+c\sigma}, \frac{1-\sigma}{2}\} < \frac{1}{6L}$.*
*(b) Theorem 1: still using $\gamma = \frac{a}{4L}$, we require $a < \frac{2}{3}$ and we get a similar (slightly smaller) expression for $\rho$ as well as for the optimal step size $\tilde{\gamma}(K) := \frac{\tilde{a}(K)}{4L}$ replacing $\gamma^*$ in the theorem:*

$$\rho = \frac{q}{n}\frac{1-\frac{3}{2}a}{1-\frac{1}{2}a} \quad and \quad \tilde{a}(K) = \frac{2K}{1+\frac{3}{2}K+\sqrt{1+K+\left(\frac{3}{2}K\right)^2}} \geq \frac{2}{3}a^*(K). \qquad (39)$$

*(c) The optimal asymptotic rate is still $\tilde{\rho} \overset{n\to\infty}{\longrightarrow} \frac{q}{n}$.*

*Proof.* Redoing all proofs with an additional factor of 2 on the RHS of Eq. (10). One can also readily verify that the ratio $\frac{\tilde{a}(K)}{a^*(K)}$ (with $a^*(K)$ defined in Eq. (18)) is a decreasing function of $K$, with value 1 for $K = 0$, and limiting value $\frac{2}{3}$ for $K \to \infty$. $\qquad\square$

## A.1 Implementation details

**Construction of the neighborhoods required by q-SAGA and $\mathcal{N}$-SAGA:** For each datapoint $i$, we want to define a neighborhood $\mathcal{N}_i$ (defined as the set of children of $i$ in a directed graph) such that for $j \in \mathcal{N}_i$, $\|\alpha_j - \beta_j\|^2 \leq \epsilon$. The approximation bounds in Section 3 show that this distance is a function of $w$, which is not known *a priori*. In order to address this issue, we used the distance between data points $\delta_{ij} := \|x_i - x_j\|$ as a surrogate. The construction of the neighborhoods then amounts to constructing a directed graph on $n$ nodes by setting the $q$ nearest points to $j$ as its *parents*.[3] This ensures that $|\{i : j \in \mathcal{N}_i\}| = q$ ($\forall j$), i.e. every $j$ has exactly $q$ parents. Note that this simple construction can yield asymmetric neighborhoods (i.e. $j \in \mathcal{N}_i \not\Rightarrow i \in \mathcal{N}_j$); $\mathcal{N}_i$ is the set of children of $i$, and does not have to be of size $q$. One could also construct a symmetric neighborhood by defining $j$ to be a child of $i$ if their distance is less than $\sqrt{\epsilon}$ (which is a symmetric relationship), where $\epsilon$ is a constant chosen such that $q \approx 20$. In practice, we did not find this construction to yield better performance (in addition to violating the uniform $q$-memorization property). Note also that the above constructions ensure that $i \in \mathcal{N}_i$.

**Growing $n$ heuristic:** For all the $q$-memorization algorithms, we used the same initialization heuristic proposed in [9, 4] for which during the first pass, datapoints are introduced one by-one, with averages computed in terms of the number datapoints processed so far (i.e. the normalization for $\bar{\alpha}$ is the number of different points seen so far instead of $n$).