[Reviews · NeurIPS 2015]

Submitted by Assigned_Reviewer_1

Quality This is a solid work that explores the trick of the gradients of neighboring data points to be similar to modify the recently proposed SAGA for SGD. The advantage of the proposed algorithm has been shown both theoretically (in Section 3) and empirically (in Section 4). However, I still think that establishing the neighborhood of the data points from the practical point of view as discussed in the paper could be an issue which could make the computational gain due to the proposed trick not as attractive as it claims.

Clarity Section 3 is not particularly easy to read through and the authors may consider adding some higher level description of the overall ideas before proceeding to the formulations and theorems.

This paper argues for its uniqueness of being able to be run in a streaming mode. This point is not clearly presented with evaluation results in the paper.

Also, there are some minor typo errors throughout the manuscript. For Eq.(3), should "i \in N_j" be "j \in N_i" instead?

Originality The trick of exploring the neighborhood is a reasonable extension of SAGA.

Significance Its significance lies on the point that the proposed enhancement is a reasonable one and is carefully studied theoretically and empirically with promising results as reported in the paper.
Summary: The paper proposed some enhanced versions of SGD by updating the gradients of also the neighboring data points with the objective to speed up the convergence. The proposed algorithm and its variants can perform better than the recently proposed SAGA.

Submitted by Assigned_Reviewer_2

NeighborhoodWatch is a existing algorithm for partial monitoring problems. Consider changing the title.

The plots are illegibly small.

Summary: The paper presents a new method for reducing variance in stochastic gradient descent, making use of stored gradients for close-by data points. The paper provides theoretical guarantees which come with a quantitative neighborhood assumption, of which it is not clearly argued that it will typically obtain. The method is evaluated on real-world data where it is shown to be competitive with the state-of-the-art competitor SAGA.

Submitted by Assigned_Reviewer_3

The paper presents an interesting improvement N-SAGA over SAGA and SVRG by considering neighboring structure during the updates of w. Though the paper provides improved results in convergence rate and experiments, it is not clear whether the new algorithm actually accomplishes any substantial benefits in a streaming and single epoch setting, as what authors claim, which is important as the work is incremental. Minor error: y-axis of Figure 3 should be test error or objective?

Summary: The paper presents an interesting improvement N-SAGA over SAGA and SVRG, SGD variants that attempt to reduce variance of w. Though the idea appears interesting but it is not clear whether this "improvement" accomplishes any significant from SAGA and SVRG.

Submitted by Assigned_Reviewer_4

The contribution of the paper is between learning and optimisation.

In machine learning we often encounter optimisation problems of the kind

\sum_i l(x_i,w) + \Omega(w) \to \min_w,

where x_i are known datapoints, w is a vector of model parameters, l is a measure of loss and \Omega is a regularisation functional. Using gradient descent may be time-consuming if the number of datapoints is large. It is therefore common to sample from datapoints as a way of approximating the true gradient.

This paper proposes to take advantage of the structure of the set of datapoints. They often cluster into a small number of neighbourhoods. New algorithms exploiting this idea are developed. Convergence guarantees and empirical analysis are provided.
Summary: The paper proposes an optimisation algorithm for machine learning utilising the fact that training points usually come from a small number of neighbourhoods.

Submitted by Assigned_Reviewer_5

-- Comments after rebuttal -- The rebuttal from the authors was concise.

However, I was not convinced about the assumption in Eq. 14 of the paper and how the

authors defended it. The authors say: "Many documents (text categorization),[..] or time series signals (speech recognition) in a training set are alike. This fact is not systematically exploited by any existing stochastic optimization method!" I don't think this is correct. In fact I think this is what statistical learning is all about,

i.e., exploit similarities in data. Also in their point (B.2) the reviewers say "Eq. 14 should not be seen as an assumption but rather as one part of a trade-off. We can make \eta arbitrarily small by restricting the size of the neighborhood." These trade-offs are very hard to do in practice. This is in fact where the crux of their method lies. It would be very interesting if the authors had a concrete answer to that.

Still, I slightly increased my rating for this paper because, judging by the rebuttal,

it seems that the authors understand the shortcomings of this paper and how to resolve them. This should allow the paper to get published if other reviewers are convinced by the rebuttal.

----

The authors study a general formulation of stochastic variance reduction (SVRG)

methods and derive a non-asymptotic analysis under strong convexity.

This work is ambitious and creative. The idea of using neighborhood

structure between data points to share information is very smart,

and looks natural only in hindsight.

However, by reading this paper I felt that it is work under progress with several

important details left unfinished. If I am mistaken I would be happy to recommend an accept.

On a high-level the paper aims to do many things at the same time, e.g.,

have a common analytic framework for SVRG methods, introduce and analyze the

idea of data neighborhoods in stochastic approximations, apply SVRG in streaming settings, etc. As a reader I felt that none of these goals were fully achieved.

Regarding the common analytic framework of SVRG methods

I struggled with assumptions in Ineq. (14) and the uniformity assumptions for the memorization

algorithms (e.g., Lines #213-215).

Why are these assumptions reasonable? I am especially skeptical of Ineq. (14) because it uniformly bounds all past and present gradients and basically assumes everything is fine with the stochastic approximation. Although I could buy such an assumption for implicit procedures (e.g.,

implicit SGD) which often have these uniform bounds,

explicit procedures as in Eq. (2) generally do not.

Regarding the streaming setting, I am confused how the authors define the streaming

setting and how they analyze and test in practice. In Line #411 the authors note: "We have found very similar results in a streaming setting, i.e., presenting data once in random

order."

I don't think this defines a streaming setting, and also seems to disagree with authors'

discussion on the streaming setting in Abstract and intro sections. My own understanding is that in streaming settings one has access to infinite

data streamed one-by-one and then discarded. One key difficulty in the analysis of SVRG procedures in streaming settings is that one has no access to unbiased estimates of the full gradient.

How do the authors cope with this difficulty? I suspect they use Lipschitz bounds as Ineq. (14)

however this is not made explicit.

Another issue here -- potentially more important -- is that

in streaming settings the estimation problem is different. In finite data sets SVRG

methods -- as well as other methods -- compute the empirical risk minimizer, e.g.,

the MLE. However, in streaming settings one estimates the true model parameters. In the former case one approximates an estimator (e.g., MLE) and in the latter case one approximates the true model parameters. Thus, in the streaming case the "linear convergence rate" of

averaging methods (mentioned by the authors in the abstract) cannot be surpassed. In the non-streaming case one can geometrically converge to the empirical minimizer, however

this minimizer is an estimator that is roughly O(1/n) away from the true model parameters. It is the nature of stochastic learning problems where the 1/n simply cannot be surpassed. This is a key issue that causes confusion in stochastic optimization from my own experience. Therefore, it would be important that the authors clarify the streaming setting they

mention in the abstract, where it appears in the analysis, and how it is being tested in experiments. Under this light, it is unclear what they mean in Line #411.

Finally, the promising idea of using neighbors of data remains underdeveloped. In Lines #138-144 the authors attempt to give practical advice on the matter but they are not specific. I am unsure how one can

efficiently define these potentially O(n^2)

neighborhood relations. The related analysis is solid using the Lipschitz conditions,

but in the experiments it was unclear how the neighborhood SVRG variants were implemented.

Other issues: - Figure 3 is not explained in main text. - Captions in figures are not informative. - Line #53, the Robbins-Monro paper is not about SGD. It is about the much more general idea of

stochastic approximations in the streaming setting.

Summary: This paper is ambitious and creative. The idea of using neighborhood

structure between data points to share information is smart, but is left underdeveloped and based on very strong/ unjustified assumptions.

Author Feedback
Author rebuttal: We would like to thank the reviewers for their insightful comments and for spotting mistakes in the manuscript.

There are a number of points that are straightforward to address in a minor revision, in particular:

(A.1) Adding precision to what we call "streaming mode". As pointed out correctly by Reviewer_3, we need to be clearer about the memory footprint of our method. Due to a tradeoff between the level of accuracy \eta in Eq. 14 and the number of past gradients that we keep, the memory needed depends on \eta and also on the data distribution. We will provide worst- and best-case examples of such data distributions to get more precise results. However, irrespective of the memory footprint, our algorithm can be run such that a fresh example is "consumed" at each iteration. While this is trivial for SGD (but without linear convergence), this mode is not supported by SAGA, SDCA, or SVRG, which are either based on the concept of batches (SVRG) or of revisiting previous data points (SAGA, SDCA).

(A.2) Presenting experimental results for the streaming data setting more clearly and with more detail on the utilized data structure (pointed out by Reviewer_3) as well as and fixing some issues with axis labels (pointed out by Reviewer_1).

(A.3) Improve the presentation and writing in light of the details in the feedback.

However, Reviewer_3 points out a few aspects of our analysis that are more fundamental, which would like to respond to.

(B.1) We agree that the assumption in Eq. 14 may be quite restrictive. How small \eta can be chosen depends on the data distribution and so does the efficiency of our algorithm. We do not claim - and maybe should be even more explicit about this fact - that our methods offers advantages for *all* data distributions. Compared to much of the work in this area, this can be seen as a weakness. However, this dependency can also be seen as a strength as it also opens up new possibilities for distribution or data-set dependent speed-ups. In many practical data sets, we have naturally arising redundancies and many training instances may be quite similar (in the input space). Many documents (text categorization), users (preference prediction), images (object recognition), or time series signals (speech recognition) in a training set are alike. This fact is not systematically exploited by any existing stochastic optimization method! So what may be "questionable" assumptions from the point-of-view of optimization theory, can well be "reasonable" when dealing with risk minimization for machine learning. We feel that our paper is innovative in exactly providing this change of perspective.

(B.2) While our method tries to exploit self-similarity properties among training instances, it does not simply fail if this assumption is not satisfied. In the practical mode of operation of N-SAGA, Eq. 14 should not be seen as an assumption but rather as one part of a trade-off. We can make \eta arbitrarily small by restricting the size of the neighborhood. Of course, as we reduce q (the average size of the neighborhoods) to q=1 in the extreme case, we also lose any speed-up relative to SAGA. So in practice, one would start with specifying a \delta (approximation accuracy) then chose \rho(\delta) and then construct neighborhoods such that Eq 14 will be approximately fulfilled (e.g. by comparing the difference in stochastic gradients).

(B.3) Empirical vs. expected risk. Of course, Reviewer_3 is absolutely right. There are two things to point out here:
(i) We are not just interested in asymptotics, but also in the transient regime (as stated in the paper), i.e. quickly get to a decent level of accuracy for the empirical risk. This accuracy may be sufficient by itself (going further might result in overfitting) or it may be a good starting point for a method that is guaranteed to converge to the empirical minimizer. Taking N-SAGA further, one could perhaps even control the neighborhoods more dynamically.
(ii) Of course, we do not claim to beat the statistical limit of O(1/n) efficiency for the generalization error :) However, maybe our use of the "streaming setting" was misleading (see above) and the writing will be improved to avoid confusion.

Conclusion: We can see that our assumption appear to be "strong" from an optimization point of view as they depend on the data-distribution. However, we think that this opens-up new possibilities for innovative stochastic gradient algorithms in machine learning, where natural data distributions may obey certain characteristics that are not yet exploited. We think that - with a few revisions and clarification mentioned - our work is a first step in this direction with a novel analysis and experimentally encouraging results. As always, there is more future research to be done here, but we feel that the paper has an end-to-end story and with a bit of "polishing" will make a solid paper that can be built upon.